# Nickel-catalyzed direct stereoselective α-allylation of ketones with non-conjugated dienes

Yi-Xuan Cao [1], Matthew D. Wodrich [2] & Nicolai Cramer [1] ✉

The development of efficient and sustainable methods for the construction of carbon-carbon bonds with the simultaneous stereoselective generation of vicinal stereogenic centers is a longstanding goal in organic chemistry. Low-valent nickel(0) complexes which promote α-functionalization of carbonyls leveraging its pro-nucleophilic character in conjunction with suitable olefin acceptors are scarce. We report a Ni(0)NHC catalyst which selectively converts ketones and non-conjugated dienes to synthetically highly valuable α-allylated products. The catalyst directly activates the α-hydrogen atom of the carbonyl substrate transferring it to the olefin acceptor. The transformation creates adjacent quaternary and tertiary stereogenic centers in a highly diastereoselective and enantioselective manner. Computational studies indicate the ability of the Ni(0)NHC catalyst to trigger a ligand-to-ligand hydrogen transfer process from the ketone α-hydrogen atom to the olefin substrate, setting the selectivity of the process. The shown selective functionalization of the α-C-H bond of carbonyl groups by the Ni(0)NHC catalyst opens up new opportunities to exploit sustainable 3d-metal catalysis for a stereoselective access to valuable chiral building blocks.

The selective formation of carbon-carbon bonds is a cornerstone of organic synthesis[1]. In this respect, the carbonyl group is particularly relevant due to its dual role. It can either react as an electrophilic component, or serve as a pronucleophile at the α-carbon in their enolized form to react with a broad range of external electrophiles[2,3]. In comparison to conventional nucleophiles or electrophiles, non-activated olefins exhibit much lower reactivity and generally require a transition-metal catalyst to engage in a reaction with carbonyl compounds[4–11]. Over the past decade, the transition from precious metals to catalysts based on earth-abundant metals made impressive progress, and especially nickel became a powerful tool due to its low-cost and unique reactivity profiles[12,13]. For instance, low-valent nickel(0) complexes promote reductive couplings between carbonyl compounds and unsaturated carbon-carbon bonds utilizing the electrophilic nature of the carbonyl group[12] (Fig. 1a). These

transformations proceed through an oxidative cyclization mechanism and subsequent hydride transfer from an external hydride source[4,14]. A variety of 1,3-dienes, allenes, enynes, alkynes can serve as the nucleophilic component leading to the formation of alkylated or alkenylated secondary and tertiary alcohols[4]. A range of chiral ligands were used for enantioselective transformations[15,16] In stark contrast, examples for the complementary α-functionalization of carbonyls with olefins under nickel(0) catalysis are scarce despite being highly desirable synthetic tools[17–20]. Unlike the above-described reductive couplings amenable to a variety of C-C π-unsaturations, the redox-neutral ketone α-functionalizations so far are very limited and only observed with selected 1,3-dienes[17,18]. The transformations require either harsh conditions and/or the addition of base or water/alcohols for a reactive catalyst system. These eventually lead to epimerization of sensitive stereocenters and thus pose challenges for diastereo-

[1]Laboratory of Asymmetric Catalysis and Synthesis, Institute of Chemical Sciences and Engineering, Ecole Polytechnique Fédérale de Lausanne (EPFL), 1015 Lausanne, Switzerland. [2]Laboratory for Computational Molecular Design, Institute of Chemical Sciences and Engineering, Ecole Polytechnique Fédérale de Lausanne (EPFL), 1015 Lausanne, Switzerland. ✉e-mail: nicolai.cramer@epfl.ch

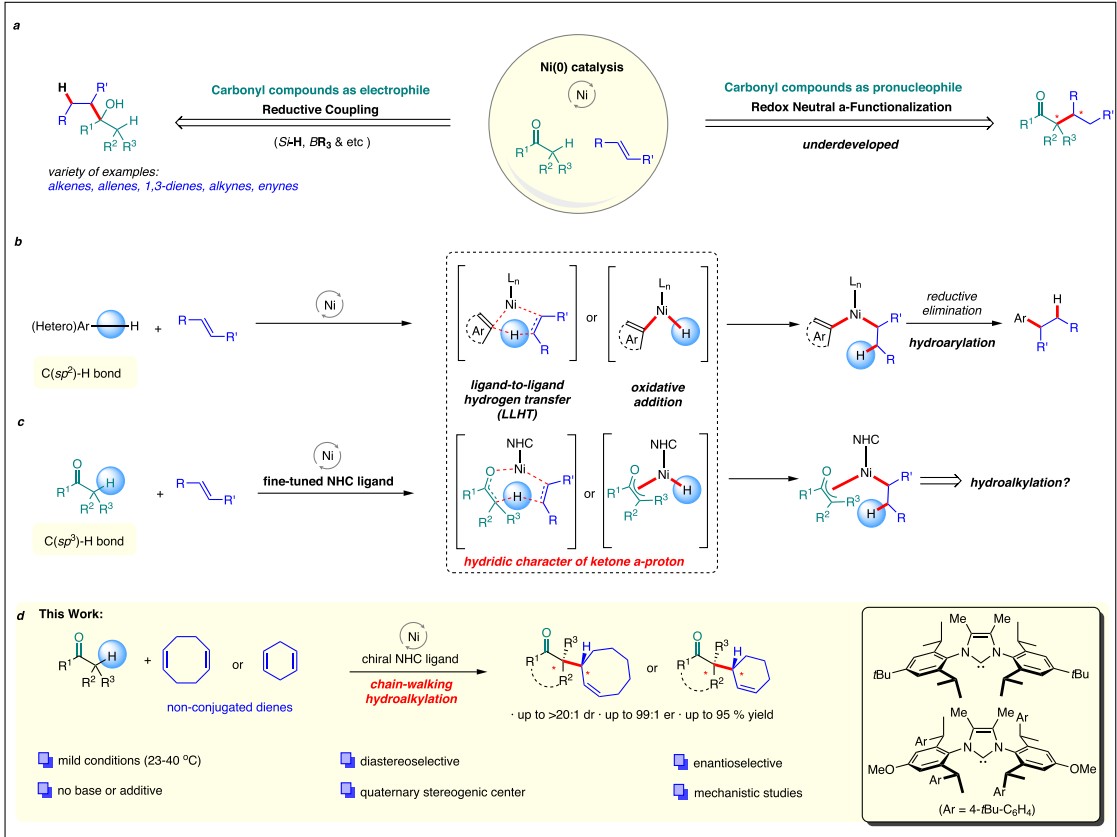

**Fig. 1 | Exploitation of ketones as pro-nucleophiles in redox neutral hydroalkylations with olefins. a** Contrasting nickel(0)-catalyzed reductive couplings of carbonyls and olefins, the complementary selective α-functionalization of carbonyls with olefins under nickel(0) catalysis remains underdeveloped. **b** Nickel(0)- catalyzed hydro-arylation of olefins **c** Hypothesized extension to Ni(0)-hydroalkylation of carbonyls and downstream engagement of intermediates in complexity-generating chain-walking processes. **d** Designer NHC enabled enantio- and diastereoselective Ni(0)-catalyzed hydro-alkylation of non-conjugated dienes.

and/or enantioselective processes. Thus, a more general approach is desirable and would be of a high synthetic value.

Transition-metal catalyzed C−H bond functionalization became an important tool to construct C−C bonds[21–24]. In this context, non-directed nickel(0)-catalyzed (hetero)arene C($sp^2$)−H activation and downstream reactions with olefins opened a successful strategy towards olefin hydro-arylations under relatively mild conditions and full atom-efficiency (Fig. 1b)[25–28]. Mechanistically, the underlying nickel-catalyzed C($sp^2$)−H bond activation process operates via a ligand-to-ligand hydrogen transfer (LLHT)[29–31] instead of the typical oxidative addition process[32]. The C($sp^2$)-C($sp^3$)-reductive elimination is considered as the rate determining step (Fig. 1b). Based on these findings, we hypothesized that an extension of this concept could help to address the α-C($sp^3$)−H bond of ketones under comparable conditions and open a new catalysis manifold for ketone α-functionalizations (Fig. 1c). This pivotal step could be either envisaged by coordination of the ketone and acceptor olefin to the nickel(0) triggering a LLHT of the carbonyl α-C($sp^3$)-H. The LLHT would involve a 7-membered transition state and lead to a favorable oxa-π-allyl nickel(II) species. Alternatively, one could formulate a C($sp^3$)−H oxidative addition forming oxa-π-allyl nickel(II) hydride. Subsequent insertion of the olefin would lead to the same alkyl nickel intermediate. The energy barrier of a straight C($sp^3$)−C($sp^3$) reductive elimination leading to direct 1,2-hydroalkylation product is higher than a corresponding C($sp^2$)−C($sp^3$) reductive elimination[33]. We thus hypothesized that the enhanced lifetime of the alkyl nickel intermediate would trigger complexity-generating downstream reactions. For instance, alkyl Ni(II) species can engage in chain-walking processes[34–42]. In the present context such increased molecular complexity comes together with challenges of controlling

regio-, diastereo-, and ultimately enantioselectivity. Capitalizing on our long-standing experience in the design and application of chiral NHC ligands for Ni(0)-catalyzed C-H functionalizations[43–47] we initiated the development of a competent catalyst system to address these challenges.

Herein, we report nickel(0) complexes equipped with a bulky chiral designer ligand enabling the fully atom-economic α-functionalization ketones with non-conjugated dienes under mild conditions (Fig. 1d). The construction of allylated quaternary centers[48] proceeds with excellent levels of diastereo- and enantioselectivity. Isolation of the NHC-Ni(0) benzene complex and mechanistic studies paired with DFT calculations support a direct LLHT transfer of the α-hydrogen atom of the ketone.

## Results

We initiated our developments of the α-allylation of ketone **1a** and 1-5 cyclooctadiene (**2a**) exploring Ni(COD)$_2$ and different bulky NHC ligands as catalyst system (Fig. 2a). Neither IPr or SIPr, the two most popular and commonly used NHC ligands for Ni(0)-catalysis provided any conversion to the expected product at ambient temperature (Entry 1-2). To our delight, by engineering the properties of the ligands, we found that SIPr^OMe[49] provided allylation product **3a** in 43% yield and with a 7:1 diastereoselectivity (Entry 3). Longer reaction time did not improve the yield (Entry 4). Acknowledging the relevance of steric bulk at the R^1-position for reactivity and complex stability, evaluation of SIPr^tBu provided **3a** in 56% yield and a slightly lower dr (6.2:1) (Entry 5). We next combined this feature with a dimethyl NHC backbone (^MeIPr^tBu). Notably, this designer NHC gave a significantly improved performance, yielding **3a** in 85% yield and 4.2:1 dr (Entry 6). The octa

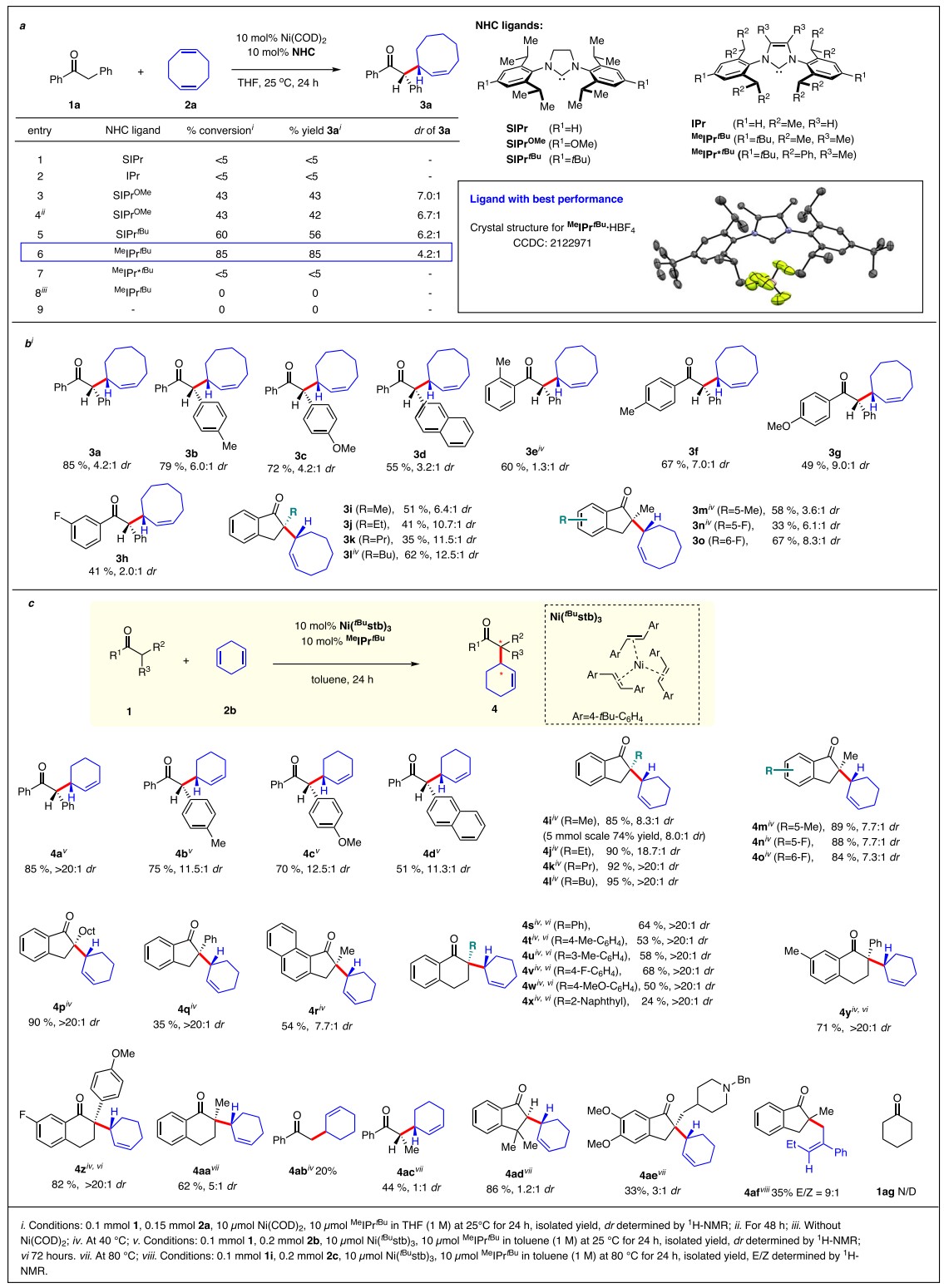

**Fig. 2 | NHC-nickel catalyzed diastereoselective α-allylation of ketones with nonconjugated dienes. a** Optimization of the diastereoselective α-allylation with 1,5-cyclooctadiene. **b** Allylations with 1,5-cyclooctadiene. **c** Allylations with 1,4-cyclohexadiene.

phenyl congener $^{Me}$IPr*$^{tBu}$ displayed poor solubility and almost no reactivity was afforded at room temperature (Entry 7). A control experiment demonstrated the essential presence of both nickel precursor and NHC ligand as no trace of product **3a** was formed in the absence of either component (Entry 8-9).

Subsequently, we investigated the scope of the hydroalkylation (Fig. 2b). In this respect, a variety of aryl benzyl ketones with different substitution pattern successfully engaged in the transformation giving the allylation products **3b**-**3h** in moderate to good yields and with a moderate to good diastereoselectivities. The use of 2-methyl indanone (**1i**) as substrate enabled construction of allylation product **3i** with an α-quaternary carbon center in good diastereoselectivity. A set of 2-substituted indanones was also subjected to the reaction conditions. Notably, increasing the size of the substituent at the 2-position further

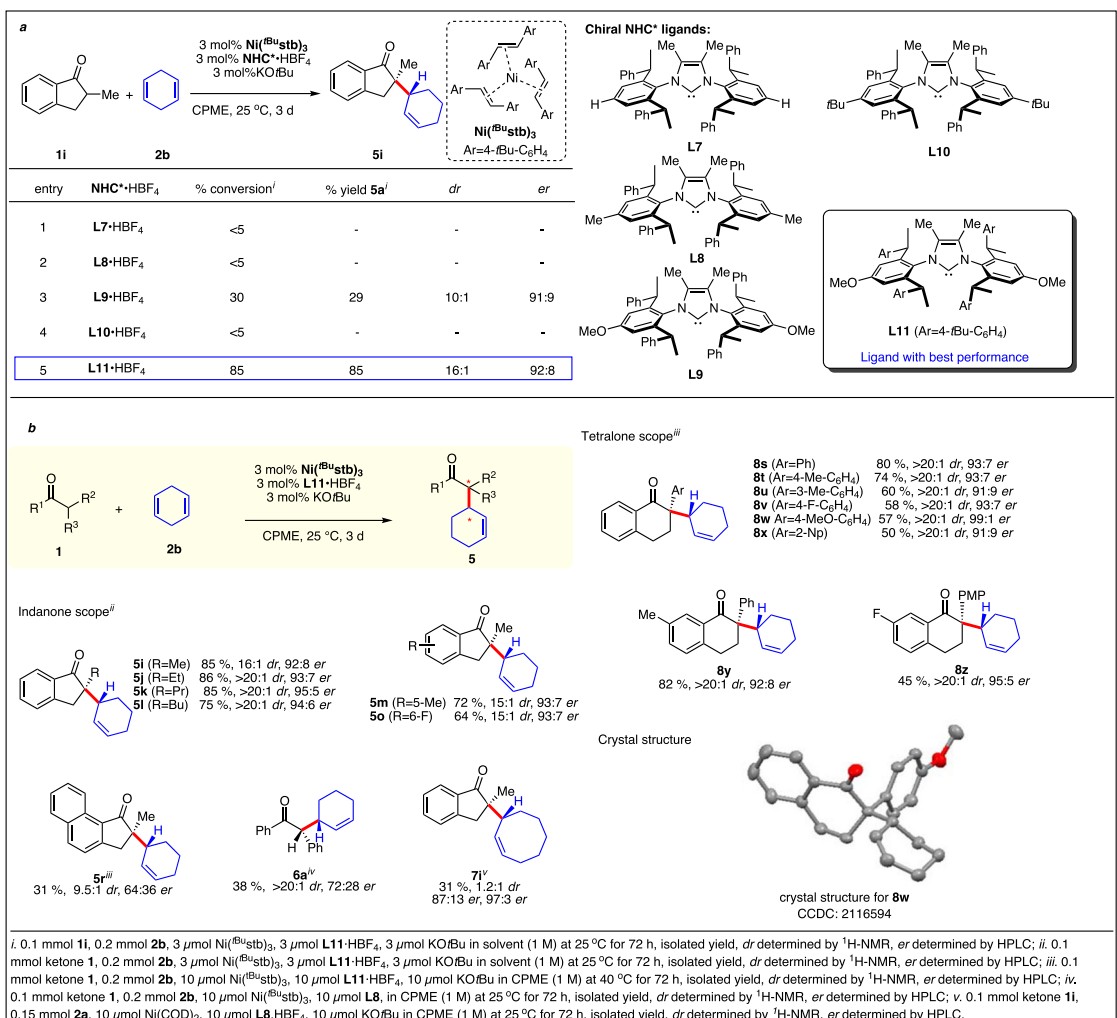

**Fig. 3 | NHC*-nickel catalyzed enantio & diastereoselective α-allylation of ketones with non-conjugated dienes. a** Optimization table of enantioselective ketone α allylation with 1,4-cyclohexadiene. **b** enantioselective allylation with non-conjugated diene.

increased the observed diastereoselectivity to up to 12.8:1 dr for **3l**. In order to expand the scope of the transformation towards further non-conjugated dienes, a COD-free nickel(0) source is required to mitigate competition between COD and the external diene reactant (Fig. 2c). Cornella's **Ni(<sup>tBu</sup>stb)₃** complex is stable, COD free and the released stilbene does not participate the reaction[50]. We identified 1,4-cyclo-hexadiene (**2b**) as attractive substrate. In this respect, aryl benzyl ketones reacted cleanly providing allylated compounds **4a–4d** with good yields and very high diastereoselectivity. Moreover, a series of 2-substituted indanones **1i-1r** successfully allowed hydroalkylation of 1,4-cyclohexadiene with generally excellent yields and excellent dia-stereoselectivities with the minor diastereomer often undetectable. Notably, hindered 2-aryl tetralones with both electronic-donating and electronic-withdrawing aryl groups reacted well and gave access to the corresponding allylated products **4s-4aa** in good yields and exquisite diastereoselectivity. Simple acyclic ketones like acetophenone or phenylethyl ketone could be allylated yielding **4ab** and **4ac** in mod-erate yield. Allylation of hindered ketone **1ad** required higher tem-perature and as a consequence a diminished diastereoselectivity of **4ad** was observed. The approved drug Donepezil (**1ae**) having a distal basic amino group reacted giving allylated derivative **4ae**. Moreover, non-conjugated 2-phenyl 1,4-pentadiene (**2c**) successfully engaged in the allylation providing products with compound (*E*)-**4af** being the most dominant regioisomer. Di-alkyl ketone **1ag** cannot deliver the desired product under reaction conditions.

With the fully atom-economic and diastereoselective process of high value, addressing the challenge of enantioselectivity of the transformation would further leverage its utility. Given the success of the above-introduced bulky designer NHC for this reaction and our ongoing activity in the development of structurally related chiral IPhEt NHCs[41,42,51–53] we evaluated a set of promising chiral NHC for the model transformation with **1b** and **2b** (Fig. 3a, SI. 4.4). IPhEt versions with the standard chiral side arm and with the dimethyl backbone such as **L1** and **L2** were incompetent ligands and no desired formation of product **5a** was observed. We next applied the gained knowledge from the diastereoselective process and synthesized ligands with 4-*t*Bu group (**L4**) or a 4-MeO group (**L3**). While a very sluggish reaction was observed with **L4, L3** provided ketone **5a** in 29% with a 30% conversion of **1a**. Pleasingly **5a** was formed with a high dr of 10:1 and a high selectivity of 91:9 er. Changing the chiral selector side arms from phenyl (**L3**) to 4-*t*Bu-phenyl (**L5**) significantly boosted the reactivity with conservation of diastereo- and enantioselectivity. Additional reaction parameters like solvent, temperature and nickel source were optimized (SI. 4.4). Finally, to bypass the need of a free carbene, in situ generation from 3 mol% **L5**·HBF₄, 3 mol% *t*BuOK and 3 mol% **Ni(<sup>tBu</sup>stb)₃** in cyclopentyl methyl ether (CPME) provided adequate reactivity and providing (*R,R*)-**5a** in 85% yield, 16:1 dr and 92:8 er. The scope of enantioselective hydroalkylation was then investigated (Fig. 3b). Bulkier substituents at the 2-position of the indanones (**5i-5l**) increased both enantioselectivities (95:5 er) and diastereoselectivities (>20:1).

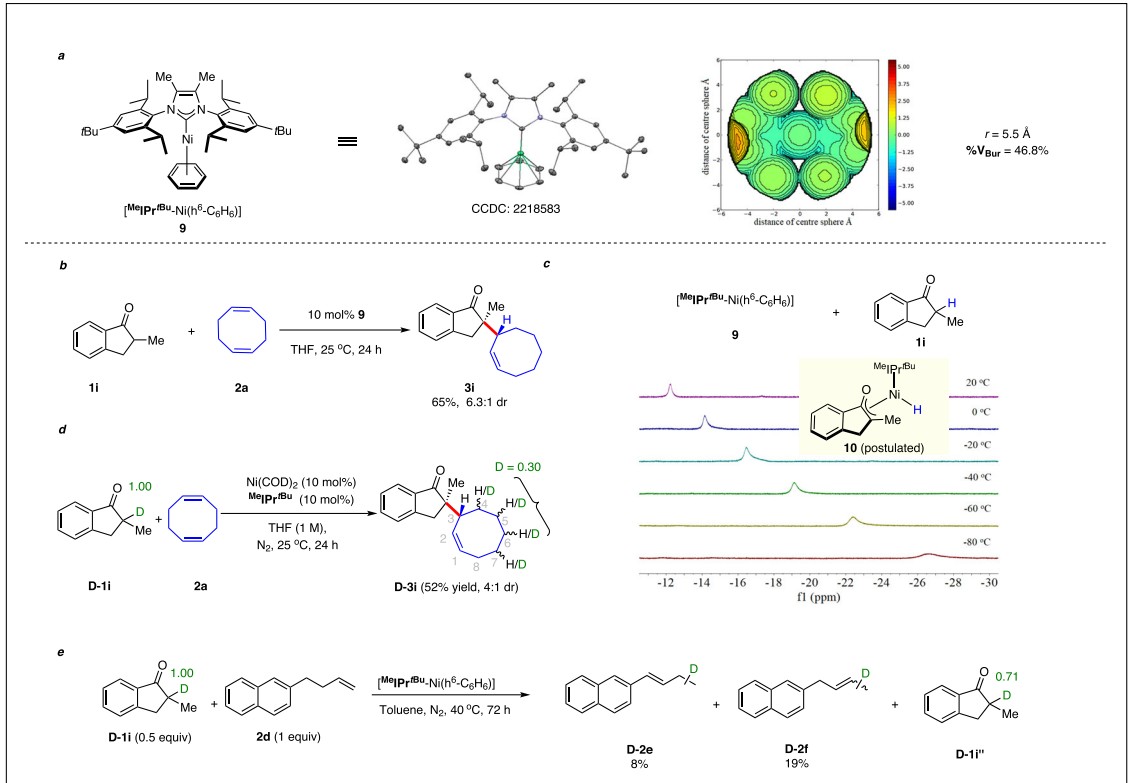

**Fig. 4 | Deuterium labeling studies. a** ORTEP diagram of [$^{Me}$IPr$^{tBu}$-Ni($\eta^6$-C$_6$H$_6$)] (thermal ellipsoids are shown at the 50% probability level, for the reason of clarity all the hydrogen atoms are omitted). Steric map and calculated buried volume. **b** Investigation of catalytic reactivity of **9**. **c** Observation of formed hydride species of reaction of **9** and **1i** via $^1$H NMR spectroscopy. **d** Deuterium scrambling experiment by monitoring reaction of **D-1i** and **2a** under standard reaction conditions. **e** Deuterium scrambling experiment by monitoring reaction of **D-1i** and **2d** under standard reaction conditions.

5-Methyl (**1 m**) or 6-fluoro indanone (**1o**) gave desired allylated products in excellent yield, diastereo- and enantioselectivity. Diminished enantioselectivity was observed for the allylation of ketone **1r** reflecting its very hindered access to the carbonyl group. Acyclic benzyl phenyl ketone **1a** with the possibility of E/Z isomers of the nickel enolate gave moderate enantioselectivity. However, the excellent diastereoselectivity of >20:1 of the process was retained. The transformation with 1,5-COD delivered **7i** with high enantioselectivities but a low diastereoselectivity. Subsequently, a series of 2-aryl substituted tetralones having electron-donating or electron-withdrawing groups reacted well yielding corresponding allylated products **8s-8z** with excellent diastereoselectivities and enantioselectivities. Notably, substrate **1w** with a PMP group increased the enantiomeric ratio to 99:1. Crystal structures of products **6a**, **8 s**, **8t**, and **8w** unequivocally allowed determination of the relative and absolute configurations formed in the allylation process.

To obtain some insights on mechanism, we synthesized and crystallized arene-bound $^{Me}$IPr$^{tBu}$-ligated nickel complex **9** (Fig. 4a)[54]. It has a buried volume of 46.8% and the influence of its p-tbutyl groups is clearly visible on the plotted steric map (SI. 5.2). Notably, **9** proved to be a competent catalyst giving **3i** in 65% yield and 6.3:1 diastereoselectivity under standard conditions. Although **9** is rather labile, its advantage is the absence of any acceptor olefin that could directly react with any potential nickel hydride intermediate, and making it impossible to observe by NMR measurements. Indeed, mixing complex **9** and ketone **1i** in $d^8$-toluene led to a new signal at -26.64 ppm at -80 °C, matching shifts of reported Ni(II) hydrides[55]. The shift of this signal is temperature dependent, an effect that was previously as well observed for iron hydrides (Fig. 4b & SI. 5.4)[56]. As expected, a mixture of complex **9** and labeled ketone **D-1i** did not form the described nickel hydride signal (SI. 5.5). Tentatively, the signal of the observed hydride

species was assigned to structure **10**, and structure **10** has been successfully located by DFT calculations (SI. Fig 6.3). Monitoring a mixture of Ni(COD)$_2$, $^{Me}$IPr$^{tBu}$ and ketone **1i** revealed that the nickel hydride signal is significantly smaller (SI. 5.6). This finding can be attributed due to the presence of two equivalents of COD able to react with nickel hydrides. Moreover, small amounts of **3i** were already detectable at 20 °C. This observation can be either explained by the reaction of COD with nickel hydride or alternatively an ongoing LLHT pathway. Reaction of D-**1i** with COD gave D-**3i** with deuterium transfer on all but the vinylic and allylic positions of the cyclooctene portion (Fig. 4d), in line with a β-hydride elimination chain-walking mechanism[33–38]. Furthermore, we observe a hydrogen/deuterium exchange between ketone D-**1i** and olefin **2d**. This finding indicates that the nickel hydride insertion or LLHT can successfully engage simple terminal mono-olefins (Fig. 4f). However, the high energy barrier of a straight C($sp^3$)–C($sp^3$) reductive elimination seams out of reach with the current catalyst system. Parallel reactions of **1b** and **D-1b** show no significant primary kinetic isotope effect, indicating that the C-H bond cleavage may not be involved in the rate-determining step (SI. 5.10). However, the observed diastereoselectivity for D-**3i** is lower than the one for **3i** (4:1 vs 6.4:1). No epimerization occurs under the reaction conditions over time. In contrast, we observed a slow erosion of the dr of **3a** dropping from 8:1 to 4.2:1 in 24 h due to presence of a remaining enolizable carbonyl group in **3a** (SI. 5.11).

To gain insights into the origin of the diastereoselectivity in the reaction between ketone **1i** and COD (**2a**) selectively forming diastereomer **3i** under Ni-$^{Me}$IPr$^{tBu}$ catalysis, we examined reaction pathways leading to diastereomers **3i** and **3i'** using density functional computations (Fig. 5a). In line with previous studies involving chain walking on nickel catalysts[35], we computed free energy profiles of the major and minor diastereomers (see SI. 6 for full computational details).

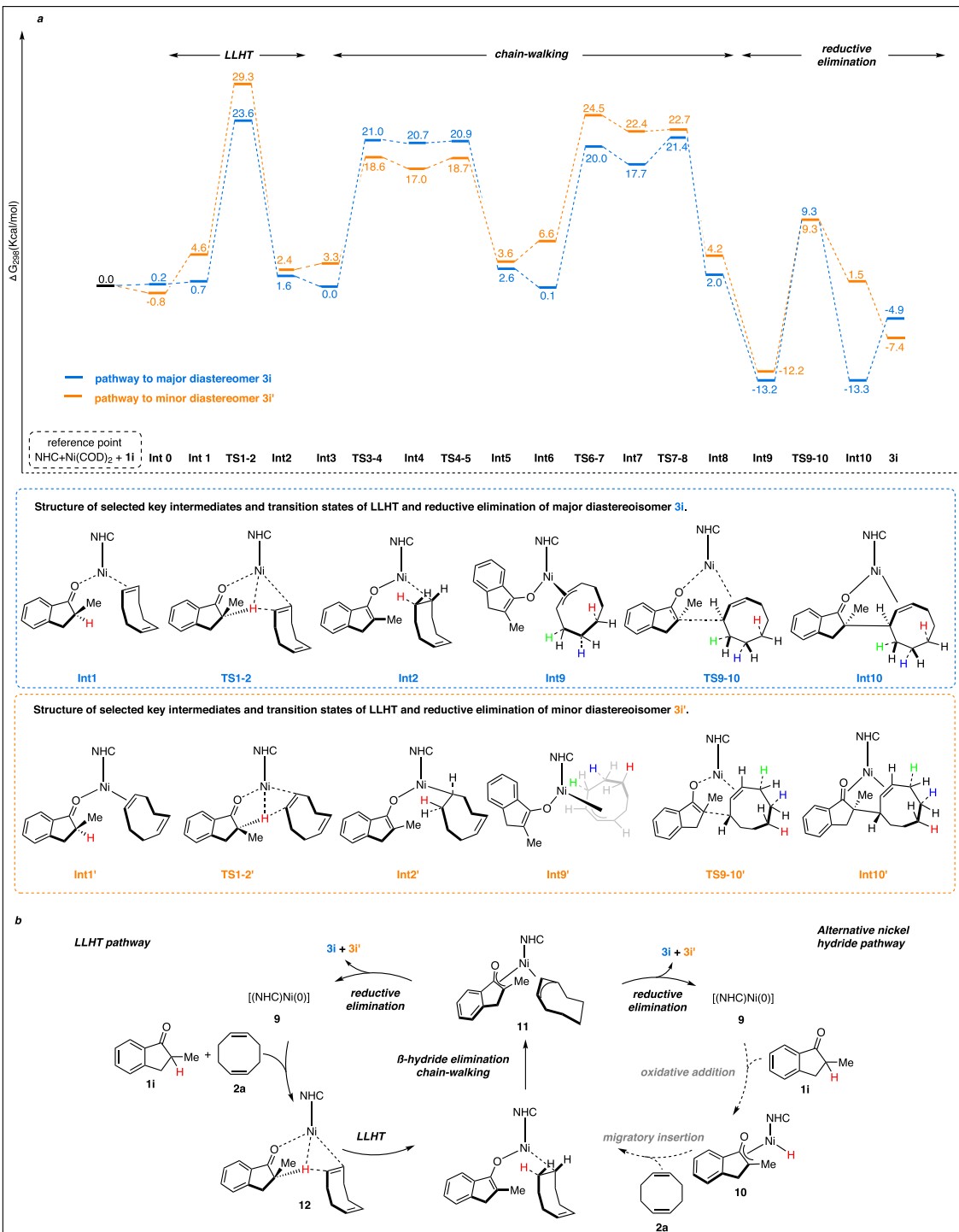

**Fig. 5 | Computational investigations if the diastereoselection. a** Computed energies for diastereoselective hydro-alkylation **b** Plausible mechanistic pathways.

We defined the reference point as Ni/NHC/COD complex and substrate ketone and began our study from **Int0**, in which **1i** and COD is bound with only one of its olefin bonds to the nickel center. The relative orientation of the ketone and the bound olefin bond of COD leads to formation of two different possible structures for **Int1** (blue vs orange pathways) that ultimately may give rise to the two experimentally observed diastereomers **3i** and **3i'**. The transition state energy of **TS1-2** that is associated with the ligand-to-ligand hydrogen transfer (LLHT) is with 23.6 kcal/mol in the typical range for the mild reaction temperatures. Notably, formation of the major diastereomer **3i** (blue pathway) is favored over the minor diastereomer **3i'** (orange pathway). As we

observed formation of nickel hydride **10** in the absence of an olefin accepter, one could invoke an alternative pathway involving a discrete nickel hydride intermediate to reach **Int2**. Although, no transition state of nickel hydride species could be located with the actual NHC nickel complex, it still remains a possible pathway, especially as there is a continuum between both mechanisms (Fig. 5b). After the initial hydride transfer onto the COD moiety, a direct reductive elimination of **Int2** is energetically disfavored under the current catalyst system. Reaction progress is initiated by an olefin chain-walking process which proceeds through a series of relatively low-energy intermediates and transition states[35] (**Int2-Int8**) to arrive at an π-allyl-bound species **Int9**.

At this stage, a final set of transition states (**TS9-10**) is associated with reductive elimination step and formation of **3i** bound to the nickel complex (**Int10**). Like for **TS1-2**, the blue pathway associated with formation of the major diastereomer is thermodynamically favored over the orange pathway of the minor diastereomer. Moreover, the significant barrier height of 22.5 kcal/mol indicates that reductive elimination could also be involved in the rate-determining step. Due to the large steric bulk of the ligand, an interconversion between the orange and blue pathways is only likely at the **Int0** stage. In such a case, the blue pathway leading to the experimentally observed major diastereomer is favorable due to the lower transition state barrier **TS1-TS2** of the LLHT.

In summary, we have developed a selective Ni(0)-catalyzed hydroalkylation of non-conjugated dienes directly involving the α-hydrogen atom of ketones *via* a LLHT or Ni-hydride formation pathway. The method converts ketones and non-conjugated dienes—both convenient, cheap and widely accessible starting materials—to highly valuable α-allylated products in excellent yields. The transformation creates adjacent quaternary and tertiary stereogenic centers in a diastereoselective manner. In addition, the design of a novel chiral bulky NHC ligand enabled the reaction to be conducted in an enantioselective manner providing the functionalized products with up to 99:1 er. Notably, the whole process is fully atom-economic. This contrasts typical carbonyl α-functionalizations with precious metals that require pre-functionalized substrates, the enolate nucleophile (ketone with base) and allyl electrophile. The mild reaction conditions are enabled by a bulky designer NHC ligand. Especially, we have observed a strong relevance of the substituent on the *para*-position of the flanking aryl groups, tentatively influencing the reductive elimination step. Surprisingly, modifications at this position were so far largely neglected in *N,N*-diaryl-based NHC ligands. Mechanistic and computational studies support that the designed Ni(0)NHC catalyst's ability to induce a hydridic character of ketone α-protons *via* LLHT processes. The alkyl nickel intermediate engages in β-hydride elimination-chain-walking sequences until the formation of π-allyl species allow reductive elimination to the observed products. We anticipate that the demonstrated facile activation of α-C-H bond of carbonyl groups by this Ni(0)NHC system can be paired with a broad range of various π-unsaturations and exploited in the future for a stereoselective and efficient general access to valuable chiral functionalized ketones by sustainable 3d-metal catalysis.

## Methods

### General procedure for products 3a–3o
In a glovebox, an oven dried screw-capped 2 mL vial was charged with a magnetic stir bar, Ni(COD)$_2$ (10 μmol), carbene ligand $^{Me}$IPr$^{tBu}$ (10 μmol), degassed (freeze pump thaw) THF (0.1 mL) was then added and the catalyst mixture was stirred at room temperature for 30 min, during which a red black solution is formed. Ketone **1** (0.1 mmol) and degassed COD (0.15 mmol) was added successively. The vial was sealed with a Teflon-lined screw cap, and the mixture was stirred in the glovebox at room temperature. After 24 h, the vial was removed from the glovebox and the reaction mixture was diluted with dichloromethane and filtered through a plug of silica gel. The crude extract was concentrated in vacuo and subjected to column chromatography 100:1 PE/EA to isolate the allylated product.

### General procedure for products 5i–5o
In a glovebox, an oven dried screw-capped 2 mL vial was charged with a magnetic stir bar, Ni($^{tBu}$stb)$_3$ (3 μmol), carbene ligand precursor **L11**·HBF$_4$ (3 μmol), KO*t*Bu (3 μmol) freshly distilled and degassed (freeze pump thaw) CPME (0.1 mL) was then added and the catalyst mixture was stirred at room temperature for 30 minutes. Then 1,4-cyclohexadiene (0.2 mmol) was added and formation of a red black

solution was observed. The mixture was stirred at room temperature for 10 minutes before ketone **1** (0.1 mmol) was added. The vial was sealed with a Teflon-lined screw cap and stirred in the glovebox at room temperature. After 72 h, the vial was removed from the glovebox and the reaction mixture was diluted with dichloromethane and filtered through a plug of silica gel. The crude extract was concentrated in vacuo and subjected to column chromatography to isolate the allylated product.

For **5r** Ni($^{tBu}$stb)$_3$ (10 μmol), carbene ligand precursor **L11**·HBF$_4$ (10 μmol), KO*t*Bu (10 μmol) was used.

For **6a** Ni($^{tBu}$stb)$_3$ (10 μmol), carbene ligand precursor **L8** (10 μmol) was used.

For **7a** Ni(COD)$_2$ (10 μmol), carbene ligand precursor **L8**·HBF$_4$ (10 μmol), KO*t*Bu (10 μmol) and diene **2a** (0.15 mmol) was used.

## Data availability
Crystallographic data for the structures reported in this article have been deposited at the Cambridge Crystallographic Data Centre under deposition numbers CCDC 2122971 ($^{Me}$IPr$^{tBu}$·HBF$_4$), CCDC 2122972 (**SIPr$^{tBu}$**·HBF$_4$), CCDC 2122973 ($^{Me}$IPr$^{*tBu}$·HBF$_4$), and CCDC 2122925 (**6a**), CCDC 2122926 (**8s**), CCDC 2122924 (**8t**), CCDC 2116594 (**8w**), CCDC 2218583 (**9**), Copies of the data can be obtained free of charge via https://www.ccdc.cam.ac.uk/structures/. All the other data including detailed experimental procedures and compound full characterization (NMR spectra, mass spectra and HPLC spectra) are available in Supplementary Information. All the raw data are available from the corresponding author upon request.

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

## Acknowledgements

This work is supported by the EPFL and the NCCR Catalysis. This publication was created as part of NCCR Catalysis (grant number 180544), a National Centre of Competence in Research funded by the Swiss National Science Foundation. We acknowledge Dr. R. Scopelliti and Dr. F. Fadaei Tirani for X-ray crystallographic analysis of compounds $^{Me}$**IPr**$^{tBu}$·HBF$_4$, **SIPr**$^{tBu}$·HBF$_4$, $^{Me}$**IPr\***$^{tBu}$·HBF$_4$, **6a**, **8s**, **8t**, **8w** and **9**. We thank Prof. C. Corminboeuf for financial support and providing computational resources.

## Author contributions

N.C and Y.-X.C designed the experiments. Y.-X.C. performed the experiments. M.D.W. performed computational studies on the reaction mechanism. All the authors analyzed the experimental & computational results and co-wrote the manuscript.

## Competing interests

The authors declare no competing interests.
