## [Peer Review File · Nature Communications]

Nickel-Catalyzed Direct Stereoselective α -Allylation of Ketones with Non-conjugated DienesEditorial Note: This manuscript has been previously reviewed at another journal that is not operating a transparent peer review scheme. This document only contains reviewer comments and rebuttal letters for versions considered at Nature Communications.

Reviewers' Comments:

Reviewer #1:

Remarks to the Author:

This manuscript is a transfer of a manuscript that I reviewed previously for Nature Chemistry, and is now under consideration at Nature Communications. The team have clearly spent some time and effort addressing the comments from the previous round of review and so I will focus here on whether these address my previous concerns.

(1) The nature of the nickel hydride complex identified from the reaction of an enolisable ketone with the Ni(0) precursor. I am satisfied that there is evidence for 'a' nickel hydride but I am not comfortable with a structure for this being postulated without sufficient evidence. If a structure for 10 can be located by DFT, then this would be sufficient to say that 10 is a reasonable proposal - this ought to then be higher in energy than Int-1, and therefore in the absence of diene the hydride is observed, but this is 'off-cycle' with respect to the catalytic reaction. If the structure of 10 is not obtainable by geometry optimisation then either (i) the level of theory is not sufficient to model this system or (ii) 10 is not a reasonable structure. In that case, I would suggest that rather than claiming a specific structure for 10, the authors simply note this as evidence of a nickel hydride without committing to a specific structure.

(As an aside: I think the text misnumbers the NHC-Ni-arene complex as 8 rather than 9 at one point - just a typo)

(2) It remains unclear what the point of the deuterium cross-over experiment is. The data are presented in two sentences and no more is said about it. There needs to be at least some nod towards why this supports the proposed mechanism. From looking at the DFT free energy profile, I think it supports the reversibility of the LLHT event (which is very slightly endergonic); i.e. Int2-d1 might then transfer back an H rather than a D instead of proceeding to the chain walking steps (the reverse of the LLHT pathway has a barrier similar to that for the onward reaction). A typo remains: "No cross-over was observed in the absence of 9 (fig. 4c)" - but fig 4c is the NMR stack plot showing evidence for a hydride.

(3) For KIE, the part " $k_H/k_D = 0.91$ " needs to be removed. The data are not of sufficient quality to support a claim of a specific value for KIE. The statement that there is no significant KIE is fine - this can be seen for the overlay of the deuterated and non-deuterated reaction profiles.

(4) The DFT data. The level of theory is satisfactory. The zero point is still not the same as that in the actual reaction - the zero point must be $[\text{Ni}(\text{COD})_2] + \text{NHC} + \text{substrates}$ because the displacement of COD from $[\text{Ni}(\text{COD})_2]$ is not always as facile as people assume: see e.g. the work of Martin ($\text{PCy}_3 + \text{Ni}(\text{COD})_2$) and Ogoshi ($\text{Ni}(\text{COD})_2 + \text{NHC}$). It is, however, reasonable to assume that free NHC forms readily under reaction conditions. This is only a minor change and should require only two straightforward geometry optimisations to minima, so I don't think this is an unreasonable request.

I feel that a comment on whether the scope of the reaction is sufficient for Nature Communications is a little beyond my area of expertise and would defer to more synthetic organic chemistry-focused reviewers on that matter.

My overall assessment is that the manuscript should be published subject to minor corrections, specifically:

- * Remove the claimed structure for 10 or at least show that the structure is stable by DFT
- * Note how the cross-over experiment supports the reaction mechanism
- * Remove any attempted quantification of the KIE from the manuscript text
- * Re-scale the free energy profile to the "true" zero - i.e. what the reaction starts from

David Nelson
(University of Strathclyde)

Reviewer #2:

Remarks to the Author:

The manuscript from Cramer and co-workers describes a Ni(0)NHC-catalyzed regio and diastereoselective α -allylation of ketones with non-conjugated dienes in a fully atom-economic manner. LLHT followed by a chain-walking-selective reductive elimination forms synthetically highly valuable α -allylated products in excellent yields under mild conditions without any additional base or additive. With a chiral bulky NHC ligand, the desired highly valuable α -C-H allylated products could be obtained in an enantioselective manner with up to 99:1 er and >20:1 dr.

The manuscript is well written, and the SI is of high-quality. Therefore I recommend publication of this manuscript in Nat. Commun. after minor revision.

- 1) Here an air-stable Ni(0) precatalyst (Ni(tBu-stb)₃) was used. Could the reaction be run at large scale? For example, 1 mmol or 5 mmol-scale?
- 2) Another significant benefit associated with the use of chiral bulky NHC ligand skeleton is that the reaction seems to proceed more efficiently (only 3 mol% cat loading) than that of using achiral NHC ligands in the first part (10 mol% cat loading). Is this a true conclusion?
- 3) For the substrate scope of ketones, only aryl ketones were shown. How about dialkyl ketones? The authors are suggested to add this information in the main text.
- 4) One or more examples of the potential synthetic application of the chiral products are suggested to be added.

Reviewer #3:

Remarks to the Author:

This manuscript was previously submitted to Nature Chemistry and most of the concerns from this referee have been addressed in the revisions. However, I still can not find the discussions on the origins of diastereoselectivity, which should be the aim of the computational studies.

--One typo in Figure 5: "major diastereomer 3i'" should be "minor diastereomer 3i'"

Raised points by Review #1:

- (1) The nature of the nickel hydride complex identified from the reaction of an enolisable ketone with the Ni(0) precursor. I am satisfied that there is evidence for 'a' nickel hydride but I am not comfortable with a structure for this being postulated without sufficient evidence. If a structure for **10** can be located by DFT, then this would be sufficient to say that **10** is a reasonable proposal - this ought to then be higher in energy than Int-1, and therefore in the absence of diene the hydride is observed, but this is 'off-cycle' with respect to the catalytic reaction. If the structure of **10** is not obtainable by geometry optimisation then either (i) the level of theory is not sufficient to model this system or (ii) **10** is not a reasonable structure. In that case, I would suggest that rather than claiming a specific structure for **10**, the authors simply note this as evidence of a nickel hydride without committing to a specific structure.

(As an aside: I think the text misnumbers the NHC-Ni-arene complex as **8** rather than **9** at one point - just a typo)

Our response: 1) The structure **10** has been successfully located by DFT calculation and these show a very low barrier for its formation. The computation details are added. (See SI Table S17 & zip file); 2) the typo has been corrected.

- (2) It remains unclear what the point of the deuterium cross-over experiment is. The data are presented in two sentences and no more is said about it. There needs to be at least some nod towards why this supports the proposed mechanism. From looking at the DFT free energy profile, I think it supports the reversibility of the LLHT event (which is very slightly endergonic); i.e. Int2-d1 might then transfer back an H rather than a D instead of proceeding to the chain walking steps (the reverse of the LLHT pathway has a barrier similar to that for the onward reaction). A typo remains: "No cross-over was observed in the absence of **9** (fig. 4c)" - but fig 4c is the NMR stack plot showing evidence for a hydride.

Our response: The deuterium cross-over experiment has been removed.

- (3) For KIE, the part " $k_H/k_D = 0.91$ " needs to be removed. The data are not of sufficient quality to support a claim of a specific value for KIE. The statement that there is no significant KIE is fine - this can be seen for the overlay of the deuterated and non-deuterated reaction profiles.

Our response: The part " $k_H/k_D = 0.91$ " has been removed.

- (4) The DFT data. The level of theory is satisfactory. The zero point is still not the same as that in the actual reaction - the zero point must be $[\text{Ni}(\text{COD})_2] + \text{NHC} + \text{substrates}$ because the displacement of COD from $[\text{Ni}(\text{COD})_2]$ is not always as facile as people assume: see e.g. the work of Martin ($\text{PCy}_3 + \text{Ni}(\text{COD})_2$) and Ogoshi ($\text{Ni}(\text{COD})_2 + \text{NHC}$). It is, however, reasonable to assume that free NHC forms readily under reaction conditions. This is only a minor change and should require only two straightforward geometry optimisations to minima, so I don't think this is an unreasonable request.

Our response: We have optimized these geometries, but disagree with the reviewer on the standpoint as reasonable zero point for the full CATALYTIC cycle and do not include them in the manuscript. The displacement of COD by a strongly coordinating ligand may or may not be fast or the free NHC may or may not form fast. This is only relevant for the first entrance in the catalytic cycle and completely irrelevant for any subsequent turn-overs. Notably, we have synthesized the well-defined NHC-Ni(0) benzene complex **9** and clearly showcased **9** as competent catalyst for the reaction in Figure 4b. Without doubt the benzene as solvent molecule dissociate immediately and the relevant starting point for catalysis is and remains **INT1**.

Raised points by Reviewer #2:

- 1) Here an air-stable Ni(0) precatalyst (Ni(tBu-stb)₃) was used. Could the reaction be run at large scale? For example, 1 mmol or 5 mmol-scale?

Our response: We have scaled the reaction to 5 mmol (with 5 mol% nickel catalyst and ligand loading). These reaction details have been added the manuscript and SI.

- 2) Another significant benefit associated with the use of chiral bulky NHC ligand skeleton is that the reaction seems to proceed more efficiently (only 3 mol% cat loading) than that of using achiral NHC ligands in the first part (10 mol% cat loading). Is this a true conclusion?

Our response: It is possible to decrease the catalyst loading for the enantioselective reactions at the expense of longer reaction times (72 h chiral cat versus 24 h achiral cat). The 3 mol% catalyst loading works well for indanone substrates, whereas tetralones require 10 mol% catalyst. Based on the all data, it is a correct conclusion that the achiral NHC ligand promotes this reaction more efficiently. All the experimental details are available in the manuscript.

- 3) For the substrate scope of ketones, only aryl ketones were shown. How about dialkyl ketones? The authors are suggested to add this information in the main text.

Our response: Under the current set of reaction conditions, cyclohexanone does not react at room temperature and undergoes aldol condensations at elevated temperatures. The information is added to the main text.

- 4) One or more examples of the potential synthetic application of the chiral products are suggested to be added.

Our response: The pattern of alpha allylated carbonyl group have been exploited numerous times for the synthesis of natural products and therefore have a proven relevance (just one recent review: *Nat. Prod. Rep.* **2018**, *35*, 559). While follow-up reactivities make sense for novel functional groups or unusual structures to provide readers knowledge of their specific intricacies and compatibilities, we do not see the relevance to include known or well-established synthetic applications besides pointing the reader to the chemistry with pertinent reviews.

Raised points by Reviewer #3:

I still can not find the discussions on the origins of diastereoselectivity, which should be the aim of the computational studies.--One typo in Figure 5: "major diastereomer 3i" should be "minor diastereomer 3i"

Our response: 1) The discussions of the "origins of diastereoselectivity" is present in the manuscript. The concerned text going together with figure 5 starts as follows:

"The relative orientation of the ketone and the bound olefin bond of COD leads to formation of two different possible structures for Int1 (blue vs orange pathways) that ultimately may give rise to the two experimentally observed diastereomers 3i and 3j'." and continues: *"Like for TS1-2, the blue pathway associated with formation of the major diastereomer is thermodynamically favored over the orange pathway of the minor diastereomer. Moreover, the significant barrier height of 22.48 kcal/mol indicates that reductive elimination could also be involved in the rate-determining step."*

2) the typo has been corrected.

Reviewers' Comments:

Reviewer #1:

Remarks to the Author:

I am satisfied with the changes to the revised manuscript.

I take on board the comments from the authors about the desire to model the system in a way that is relevant to the catalytic cycle, and the reactivity of NHC-Ni-benzene complexes where the benzene ligand is known to be loosely bound.

Reviewer #2:

Remarks to the Author:

I endorsed publication in the last round, but I again appreciate the authors addressing my comments (as well as good questions from the other referee).